# From Whole-Body to Abdomen: Streamlined Segmentation of Organs and Tumors via Semi-Supervised Learning and Efficient Coarse-to-Fine Inference

Shoujin Huang[1], Huaishui Yang[1], Lifeng Mei[1], Tan Zhang[1],
Shaojun Liu[1], and Mengye Lyu[1]

Shenzhen Technology University
`lvmengye@sztu.edu.cn`

**Abstract.** Precise and automated segmentation of abdominal organs and tumors is an important research area of medical image analysis. This domain faces three key challenges: the presence of partially labeled training data that can mislead model training, the variable morphologies of tumors complicating the segmentation process, and the computationally demanding nature of inference in whole/half-body CT scans. In our study, we leverage advanced techniques to generate pseudo-labels, thereby adequately addressing the limitations of partially annotated datasets in a semi-supervised manner. Furthermore, we introduce a novel perspective that allows the segmentation of whole/half-body CT scans to be streamlined into focused abdominal segmentation. To achieve this, we re-engineered the nnU-Net V2 inference engine to incorporate a coarse-to-fine strategy, leading to a remarkable $15\times$ speed-up by eliminating extraneous regions. The mean under the GPU memory-time curve is 7918 Mb. Our approach yields a mean Dice Similarity Coefficient (DSC) of 90.75/47.95 and a Normalized Surface Dice (NSD) of 95.54/40.16 for organ and tumor segmentation, respectively, in the FLARE 2023 validation dataset. Importantly, our method accomplishes these results with an average processing time of only 27.47 seconds per case.

**Keywords:** FLARE2023 · Segmentation

## 1 Introduction

Automated and precise segmentation of abdominal organs and tumors is crucial for a wide range of medical applications, including computer-assisted diagnosis and biomarker measurement systems. The growing need for automated segmentation in abdominal medical imaging highlights its essential role in facilitating accurate diagnoses, surgical planning, and disease localization. This area faces three key challenges. First, the existence of partially labeled training data complicates model learning. Second, the varied morphologies of tumors present difficulties for accurate segmentation. Third, the high voxel count in whole or

half-body CT scans requires significant computational resources, leading to prolonged inference times.

Previously, Z. Huang et al. employed big nnU-Net models to generate effective pseudo-labels, which were then provided to small nnU-Net models for learning [8]. F. Zhang et al. utilized model distillation techniques along with unlabeled data [20], achieving significant improvement on segmentation accuracy compared to full-supervised models. Both of their approaches underscore the potential of semi-supervised pseudo-labeling methods in enhancing model robustness. On the other hand, S. Huang et al. focused on optimizing processing time through GPU-based re-implementation of several frequent operations, leading to a substantial increase in inference speed [7].

In this paper, we utilize the FLARE 2022 winner model[9] to generate a large set of pseudo-labels for the issues of partially labeled training data. These pseudo-labels play a crucial role in augmenting annotations within both the partially labeled and unlabeled datasets. Subsequently, they are employed for training nnU-Net models. Furthermore, we introduce a new approach wherein mask calculations, performed via patch slide window prediction for whole/half-body CT scans, can be essentially refocused solely on abdominal region segmentation. This refocus is justified by the fact that only the abdominal regions containing the target organs require high-precision inference; all other regions may be selectively excluded or omitted entirely. As a result, we have modified the nnU-Net V2 inference engine to include a coarse-to-fine strategy. This adaptation dramatically reduces the need for processing extraneous regions (such as the head and feet), thereby achieving precise abdominal segmentation along with an impressive 15-fold increase in processing speed.

Our main contributions are summarized as follows:

- We introduce a novel perspective that allows the segmentation of whole/half-body CT scans to be streamlined into focused abdominal segmentation. This refines the computational scope and significantly alleviates the need for extensive computational resources.
- By incorporating a coarse-to-fine strategy into the nnU-Net V2 inference engine, we achieve a remarkable 15-fold acceleration in processing speed without compromising on segmentation accuracy.
- Through rigorous experimentation, we validate the superior performance and efficiency of our innovative framework, establishing it as a robust method for precise segmentation of both organs and tumors in abdominal CT scans.

## 2   Method

### 2.1   Preprocessing

Following [10], we implement the following preprocessing steps:

- Crop individual scans to the non-zero region.
- Apply global dataset intensity percentile clipping and z-score normalization using global foreground mean and standard deviation.

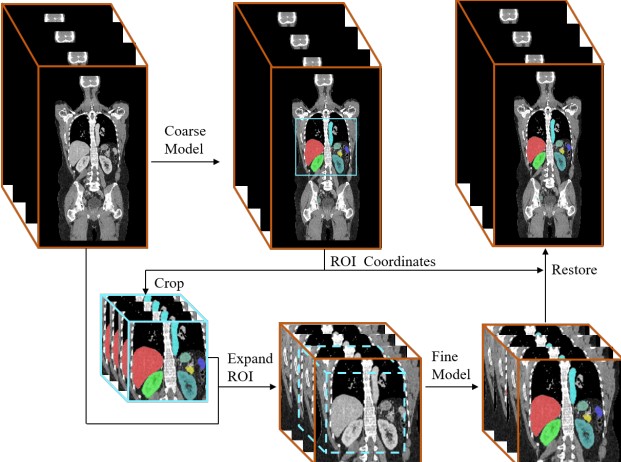

**Fig. 1.** The proposed coarse-to-fine segmentation streamline. The inference from the coarse model is used for the fine model to focus on the abdominal ROI. Crop means cutting the approximate position of the organs/tumors from the original image, and Restore means placing the prediction back to the position before cropping.

– Train the coarse model using data with a spacing of $(2.5, 0.79, 0.79)$, and the fine model with a spacing of $(0.5, 0.79, 0.79)$.

We utilize intensity percentile clipping and normalization based on the global foreground mean and standard deviation. These preprocessing steps are essential, as CT scan values represent physical properties that must be preserved in the processed data.

## 2.2 Streamline

In terms of inference stage, as a core part of this study, we introduce a segmentation streamline as illustrated in Fig. 1. Initially, we employ the coarse model with a step size of 1 to obtain approximate segmentation results from the input CT scan. Subsequently, we extract the coordinates of the abdomen region of interest (ROI) based on the coarse segmentation. The ROI box is then expanded by 20 mm in every direction to ensure that the organ is fully encompassed within it. Following this, we crop the corresponding area and perform inference using the fine model with a step value of 0.5. Finally, we restore the inference results to their original cropped area using the ROI coordinates. Note that we perform all interpolation operations using GPU device instead of CPU device.

## 2.3 Coarse and Fine Models

The configuration details of our coarse and fine segmentation models are outlined in Table 1. Both models are built upon the U-Net architecture [16] from nnU-Net

**Table 1.** Coarse and fine model implementation

| Settings | Coarse Model | Fine Model |
|---|---|---|
| Channels in the first stage | 24 | 32 |
| Convolution number per stage | 2 | 3 |
| Downsampling times | 5 | 5 |
| Step size | 1 | 0.5 |
| Input patch size | (96,160,128) | (96,160,128) |
| Input spacing | (2.5,0.79,0.79) | (0.8,0.79,0.79) |

V2 [10] and include downsampling and upsampling layers. The downsampling layers are responsible for reducing the scale of features and expanding feature channels, while the upsampling layers upscale the downsampled features. These are then concatenated with skip connections before undergoing convolution to obtain the final feature maps. It is noteworthy that the primary objective of the coarse segmentation model is to swiftly identify ROI with less emphasis on segmentation accuracy. Consequently, we have reduced the parameters of the coarse model and optimized step sizes to enhance speed for ROI extraction. In contrast, the fine segmentation model demands higher precision. Additionally, we employed a loss function that combines Dice Loss and Cross-Entropy Loss.

## 3    Experiments

### 3.1    Dataset and Evaluation Measures

The FLARE 2023 challenge extends previous FLARE 2021-2022 [12,13], aiming to promote the development of foundation models for abdominal disease analysis. The segmentation targets encompass 13 organs and various abdominal lesions. The training dataset has been curated from more than 30 medical centers, with licensing permissions, and includes sources such as TCIA [2], LiTS [1], MSD [17], KiTS [5,6], autoPET [4,3], TotalSegmentator [18], and AbdomenCT-1K [14]. The training set comprises 4,000 abdominal CT scans, of which 2,200 have partial labels and 1,800 lack labels. The validation and testing sets include 100 and 400 CT scans, respectively, covering various types of abdominal cancer such as liver, kidney, pancreas, colon, and gastric cancer. Organ annotations were performed using ITK-SNAP [19], nnU-Net [10], and MedSAM [11].

The evaluation metrics include two accuracy measures: the Dice Similarity Coefficient (DSC) and the Normalized Surface Dice (NSD), as well as two efficiency measures: running time and area under the GPU memory-time curve. These metrics collectively contribute to the overall ranking computation. Additionally, the running time and GPU memory consumption are evaluated within tolerances of 15 seconds and 4 GB, respectively.

### 3.2    Implementation Details

**Environment settings** The development environments and requirements are presented in Table 2.

**Table 2.** Development environments and requirements.

| | |
|---|---|
| System | Ubuntu 22.04 |
| CPU | AMD Ryzen 9 5900X 12-Core Processor |
| RAM | 4×32GB |
| GPU (number and type) | Two NVIDIA GeForce RTX 3090 24G |
| CUDA version | 12.2 |
| Programming language | Python 3.9.7 |
| Deep learning framework | torch 2.0, torchvision 0.2.2 |
| Code | |

**Training protocols** Regarding the coarse model, it is unnecessary distinguish tumor segmentation results. Hence, we exclusively utilized the 13 available organ labels in the dataset to train this model. In contrast, for the fine model, we also utilized the partially labeled data with pseudo-labels generated by Z. Huang [9]. Subsequently, we trained the fine model through the nnU-Net V2 framework. During the initial training phase, 80% of the labels were used for training, while the remaining 20% were reserved for validation. In the final training phase, we selected the fine model with the lowest validation loss and performed a fine-tuning process using the entire pseudo-label dataset. Additionally, we adjusted the initial learning rate to 1e-4 and disabled data augmentation during this fine-tuning procedure.

## 4 Results and Discussion

### 4.1 Quantitative Results on Validation Set

The public validation represents the performance on the 50 validation cases with ground truth. The online validation corresponds to the leaderboard results. The testing results will be released during MICCAI 2023. As shown in Table 3, we provide both the mean scores and standard deviations.

### 4.2 Qualitative Results on Validation Set

Fig. 2 displays two examples with good segmentation results and two examples with poor segmentation results in the validation set. In the case of Case#43 and Case#77, our method successfully segments all organs, achieving high DSC scores. In the case of Case#32 and Case#87, our method also performs well on large organs with clear boundaries (e.g., liver and stomach) but struggles with tumor segmentation. Furthermore, after the utilization of pseudo-labels, kidney and spleen segmentation significantly improves. The experimental results underscore the capacity of pseudo-labels to enhance the accuracy of our algorithm.

### 4.3 Segmentation Efficiency on Validation Set

Table 4 presents the efficiency measures of inference for various samples. Notably, as the number of slices increases, the time required for some of our samples

**Table 3.** Quantitative evaluation results.

| Target | Public Validation | | Online Validation | | Testing | |
|---|---|---|---|---|---|---|
| | DSC(%) | NSD(%) | DSC(%) | NSD(%) | DSC(%) | NSD (%) |
| Liver | 98.22 ± 1.32 | 98.41 ± 3.47 | 98.29 | 98.59 | 96.06 | 96.42 |
| Right Kidney | 95.51 ± 7.70 | 95.45± 8.87 | 94.45 | 94.71 | 94.52 | 95.07 |
| Spleen | 98.00 ± 1.17 | 99.03 ± 1.80 | 98.15 | 99.20 | 96.66 | 97.80 |
| Pancreas | 87.56 ± 6.48 | 97.08 ± 4.22 | 86.72 | 96.40 | 89.69 | 96.11 |
| Aorta | 96.62 ± 2.40 | 98.56 ± 3.02 | 96.66 | 98.56 | 96.64 | 98.86 |
| Inferior vena cava | 94.56 ± 3.05 | 96.37 ± 3.82 | 94.06 | 95.48 | 93.91 | 95.92 |
| Right adrenal gland | 85.56 ± 5.36 | 96.88 ± 2.63 | 84.39 | 96.23 | 82.09 | 94.22 |
| Left adrenal gland | 83.46 ± 6.38 | 95.60 ± 3.74 | 82.38 | 94.35 | 81.37 | 93.61 |
| Gallbladder | 85.47 ± 18.99 | 86.17 ± 20.20 | 85.26 | 85.68 | 81.95 | 84.51 |
| Esophagus | 83.71 ± 15.83 | 92.76 ± 15.23 | 84.40 | 94.10 | 90.51 | 98.10 |
| Stomach | 93.79 ± 3.66 | 96.90 ± 4.77 | 94.22 | 97.46 | 93.45 | 97.11 |
| Duodenum | 83.34 ± 7.03 | 95.00 ± 5.05 | 83.73 | 95.45 | 86.26 | 96.77 |
| Left kidney | 93.95 ± 11.13 | 93.81 ± 12.15 | 93.80 | 94.07 | 93.50 | 94.37 |
| Tumor | 47.95 ± 34.47 | 40.16 ± 32.39 | 43.12 | 36.87 | 40.18 | 30.85 |
| Average | 90.75 ± 6.96 | 95.54 ± 6.84 | 90.50 | 95.41 | 90.33 | 95.14 |

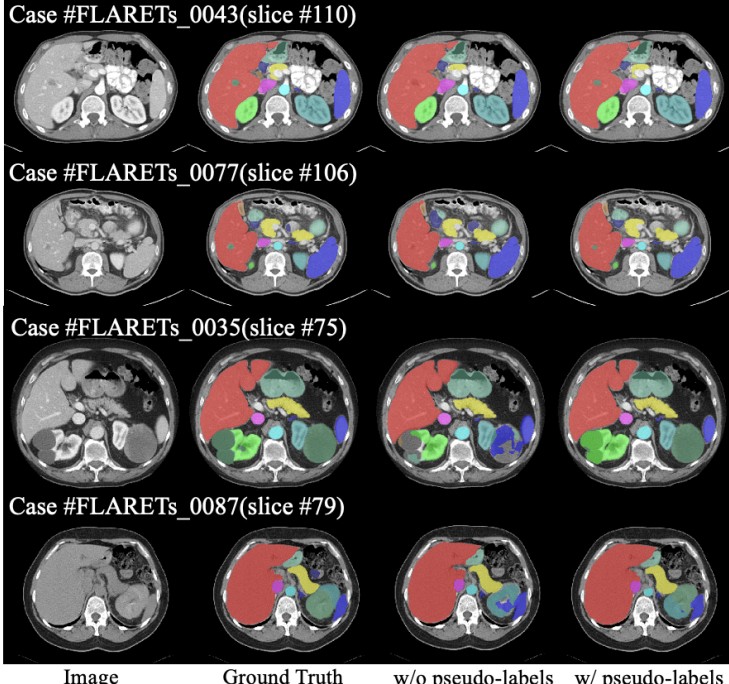

**Fig. 2.** Two examples with good segmentation and two examples with poor segmentation in the validation. The first column displays the image, the second column shows the ground truth, the third column presents the prediction by the model trained with labeled data but without tumor annotations, and the fourth column shows the prediction by the model trained with pseudo-labels.

does not increase proportionally. For instance, when comparing Case#0048 and Case#0063, it becomes evident that Case#0048 requires less time despite having a greater number of slices. This discrepancy is primarily due to the incorrect computation of the excessively large ROI box in Case#0063, resulting in an extended duration for the second step involving the fine model. In response to time constraints during the competition, we opted not to implement post-processing steps aimed at enhancing the quality of the coarse segmentation to maintain robustness in the coarse process. However, the successful handling of high-slice samples like Case#0099 and Case#0048 demonstrates that we can effectively treat large-slice samples as if they were smaller ones during the inference process. This further underscores the effectiveness and superiority of our proposed coarse-to-fine segmentation streamline.

### 4.4   Ablation Study

In this section, we conducted experiments to investigate the improvement in robustness resulting from the utilization of pseudo-labels, as well as the time reduction achieved through our proposed coarse-to-fine acceleration strategy. Please note that our ablation study experiments were conducted within the environments and requirements presented in Table 2, using the public validation dataset.

**Table 4.** Quantitative evaluation of segmentation efficiency in terms of the running time and GPU memory consumption. Total GPU denotes the area under GPU memory-time curve. Evaluation GPU platform: NVIDIA QUADRO RTX5000 (16G).

| Case ID | Image Size | Running Time (s) | Max GPU (MB) | Total GPU (MB) |
|---------|------------|------------------|--------------|----------------|
| 0001 | (512, 512, 55) | 22.75 | 5596 | 53528 |
| 0051 | (512, 512, 100) | 33.25 | 11448 | 152572 |
| 0017 | (512, 512, 150) | 35.91 | 11420 | 146076 |
| 0019 | (512, 512, 215) | 29.89 | 5472 | 52459 |
| 0099 | (512, 512, 334) | 24.75 | 5868 | 66231 |
| 0063 | (512, 512, 448) | 34.87 | 11396 | 133175 |
| 0048 | (512, 512, 499) | 30.03 | 7328 | 85168 |
| 0029 | (512, 512, 554) | 50.34 | 16170 | 255586 |

**Effect of pseudo-labels.** In this experiment, labeled data refers to 597 samples that possess organ annotations but lack tumor annotations, and pseudo-labels denote annotations generated using Z. Huang et al.'s method [9] to complete and rectify the partially labeled data. Table 5 illustrates the impact of incorporating pseudo-labels alongside labeled data during the training of the organ segmentation model. It is evident that the inclusion of pseudo-labels results in a noticeable improvement in model accuracy.

**Table 5.** Ablation study on pseudo-labels data effection.

| Variant | Organ | | Tumor | |
|---|---|---|---|---|
| | DSC(%) | NSD(%) | DSC(%) | NSD(%) |
| *nnU-Net V2+labeled data* | 88.66 | 94.09 | - | - |
| *nnU-Net V2+pseudo-labels data* | 90.20 | 95.15 | 48.10 | 40.21 |
| *ours+labeled data* | 89.67 | 94.92 | - | - |
| *ours+pseudo-labels data* | 90.75 | 95.54 | 47.95 | 40.16 |

**Effect of coarse-to-fine design.** Table. 6 provides an overview of the time consumption achieved by our proposed method in various variants. *nnU-Net V2* refers to the original inference engine of nnU-Net V2 without any modifications, and *w/o coarse model* indicates using our proposed method but without the coarse model for ROI area computation yet with the GPU-based interpolation. From the experimental results, it becomes evident that when tested with identical model weights, our proposed engine framework significantly outperforms plain *nnU-Net V2* in terms of accuracy, exhibiting a notable improvement of 0.55/0.39 in organ segmentation. In terms of time efficiency, our method also surpasses the original inference of nnU-Net V2, achieving an impressive 15× acceleration.

**Table 6.** Ablation study on time analysis.

| Variant | Organ | | Tumor | | Consume time |
|---|---|---|---|---|---|
| | DSC(%) | NSD(%) | DSC(%) | NSD(%) | seconds |
| *nnU-Net V2* | 90.20 | 95.15 | 48.10 | 40.21 | 6959.82 |
| *w/o coarse model* | 89.22 | 93.78 | 47.52 | 39.37 | 1090.46 |
| *ours* | 90.75 | 95.54 | 47.95 | 40.16 | 467.96 |

### 4.5 Results on Final Testing Set

Our approach demonstrates a mean Dice Similarity Coefficient (DSC) of 90.33/95.14 and a Normalized Surface Dice (NSD) of 40.18/30.85 for organ and tumor segmentation, respectively, in the FLARE 2023 final testing set. Furthermore, the average processing time is 20.26s, with GPU memory utilization at 54,842 MB.

### 4.6 Limitation and Future Work

The current study has a few limitations. Firstly, we were unable to introduce post-processing analysis after coarse and fine segmentation to enhance the algorithm's robustness due to competition time constraints. Secondly, we struggled to strike a balance between model accuracy and efficiency, prolonging inference times even with the utilization of proposed framework. Finally, directly performing image interpolation on the GPU could potentially lead to insufficient GPU

memory, thereby causing program crashes. In our future work, we will incorporate GPU-based post-processing after segmentation. Additionally, we aim to implement image interpolation efficiently on the GPU platform with low resource consumption.

## 5   Conclusion

Our proposed novel perspective and inference framework have proven effective, and the incorporation of pseudo-labels has been shown to enhance model robustness. Initially, we introduced the concept of simplifying the segmentation of whole/half-body CT scans into abdominal segmentation. Building upon this concept, we restructured the inference framework based on nnU-Net V2 and employed a coarse-to-fine segmentation approach. Experimental results demonstrate that our novel approach achieves a remarkable $15\times$ speedup in segmentation compared to the original nnU-Net V2 inference engine while preserving tumor segmentation accuracy to a significant extent. Furthermore, there is a noticeable improvement in organ segmentation accuracy.

**Acknowledgements** The authors of this paper declare that the segmentation method they implemented for participation in the FLARE 2023 challenge has not used any pre-trained models nor additional datasets other than those provided by the organizers. The proposed solution is fully automatic without any manual intervention. We thank all the data owners for making the CT scans publicly available and CodaLab [15] for hosting the challenge platform. This work was supported in part by the National Natural Science Foundation of China under Grant 62101348, the Shenzhen Higher Education Stable Support Program under Grant 20220716111838002, and the Natural Science Foundation of Top Talent of Shenzhen Technology University under Grants 20200208 and GDRC202117.

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

**Table 7.** Checklist Table. Please fill out this checklist table in the answer column.

| Requirements | Answer |
|---|---|
| A meaningful title | Yes |
| The number of authors ($\leq 6$) | 6 |
| Author affiliations, Email, and ORCID | Yes |
| Corresponding author is marked | Yes |
| Validation scores are presented in the abstract | Yes |
| Introduction includes at least three parts: background, related work, and motivation | Yes |
| A pipeline/network figure is provided | Fig. 1 |
| Pre-processing | 2 |
| Strategies to use the partial label | 5 |
| Strategies to use the unlabeled images. | 5 |
| Strategies to improve model inference | 2,w3 |
| Post-processing | No |
| Dataset and evaluation metric section is presented | 4 |
| Environment setting table is provided | Table. 2 |
| Training protocol table is provided | Table. 1 |
| Ablation study | 7,8 |
| Efficiency evaluation results are provided | Table. 3 |
| Visualized segmentation example is provided | 6 |
| Limitation and future work are presented | Yes |
| Reference format is consistent. | Yes |