# OpenReview forum: "From Whole-Body to Abdomen: Streamlined Segmentation of Organs and Tumors via Semi-Supervised Learning and Efficient Coarse-to-Fine Inference"
_MICCAI.org/2023/FLARE — Submitted to FLARE 2023_

### Official Review · Reviewer_2JeH · 2023-10-04
**From Whole-Body to Abdomen: Streamlined Segmentation of Organs and Tumors via Semi-Supervised Learning and Efficient Coarse-to-Fine Inference**

**Rating:** 7
**Confidence:** 4

**Review:**

This paper introduces an experiment based on nnU-Net V2, and according to the proposed coarse-to-fine segmentation streamline, the whole abdomen is roughly segmented, more precise segmentation is performed, and ROI Coordinates are used to restore. It is interesting to note that the coarse model is optimized for faster ROI extraction, while the fine model has larger parameters and step sizes for higher accuracy. The results are impressive with very high organ and tumor DSC, NSD.

---

### Official Review · Reviewer_SspE · 2023-10-19
**two-stage coarse-to-fine nnUnet V2 achieve good result**

**Rating:** 6
**Confidence:** 4

**Review:**

Pros：The author raised three challenges that needed to be addressed, including partial annotations, various tumor morphologies, and a large number of pixel calculations. By generating pseudo label and optimizing the segmentation strategy for abdominal region using nnU-Net V2, they achieved a significant improvement in accuracy and speed from coarse to fine segmentation. The method's introduction is clear, and the content is comprehensive.

Cons：There are a few areas where clarity could be improved: 1.Why is there significant fluctuation in GPU utilization at the maximum level? 2.Since the competition is about pan-tumor segmentation, does focusing only on abdominal region segmentation affect the recognition of tumors in other areas? 3.The increase in accuracy after incorporating pseudo-labels is not very significant; are there more effective strategies or methods available?

---

### Official Review · Reviewer_Am2F · 2023-10-25
**From Whole-Body to Abdomen: Streamlined Segmentation of Organs and Tumors via Semi-Supervised Learning and Efficient Coarse-to-Fine Inference**

**Rating:** 10
**Confidence:** 5

**Review:**

Please include the area under the GPU memory-time curve on the public validation set.

---

### Official Review · Reviewer_oFd1 · 2023-10-25

**Rating:** 5
**Confidence:** 5

**Review:**

Abstract: please include and the average running time and area under GPU memory-time curve.
Sec 2.2: Please introduce your loss function.
Sec 2.2: Please introduce your strategies to deal with the partial labels.
Sec 2.2: Please introduce your strategies to use the 1800 unlabeled images. If you don’t use them, please explicitly say "Unlabeled images were not used."
Sec 2.2: Please introduce your strategies to improve inference speed and reduce resource consumption.
Please give a meaningful title to the 3rd column in Fig. 2 instead of using ablation study.

---

### Official Review · Reviewer_3ScQ · 2023-10-25

**Rating:** 6
**Confidence:** 5

**Review:**

1. The average running time and area under GPU memory-time curve are missing in the Abstract section.
2. Please introduce your strategies to deal with the partial labels.
3. Please introduce your strategies to use the 1800 unlabeled images.
4. Please introduce your strategies to improve inference speed and reduce resource consumption.

---

### Decision · Program_Chairs · 2023-10-25

Accept